# Associations of Exercise Habits in Adolescence and Old Age with Risk of Osteoporosis in Older Adults: The Bunkyo Health Study

**DOI:** 10.3390/jcm10245968

**Published:** 2021-12-19

**Authors:** Hikaru Otsuka, Hiroki Tabata, Huicong Shi, Hideyoshi Kaga, Yuki Someya, Abudurezake Abulaiti, Hitoshi Naito, Futaba Umemura, Saori Kakehi, Muneaki Ishijima, Ryuzo Kawamori, Hirotaka Watada, Yoshifumi Tamura

**Affiliations:** 1Department of Sports Medicine and Sportology, Graduate School of Medicine, Juntendo University, 2-1-1 Hongo, Bunkyo-ku, Tokyo 113-8421, Japan; h.otsuka.eq@juntendo.ac.jp (H.O.); h.shi.li@juntendo.ac.jp (H.S.); yksomeya@juntendo.ac.jp (Y.S.); f.umemura.wg@juntendo.ac.jp (F.U.); ishijima@juntendo.ac.jp (M.I.); kawamori@juntendo.ac.jp (R.K.); 2Sportology Center, Graduate School of Medicine, Juntendo University, 2-1-1 Hongo, Bunkyo-ku, Tokyo 113-8421, Japan; abudoure@juntendo.ac.jp (A.A.); skakei@juntendo.ac.jp (S.K.); hwatada@juntendo.ac.jp (H.W.); 3Metabolism and Endocrinology, Graduate School of Medicine, Juntendo University, 2-1-1 Hongo, Bunkyo-ku, Tokyo 113-8421, Japan; hkaga@juntendo.ac.jp (H.K.); h-naito@juntendo.ac.jp (H.N.); 4Graduate School of Health and Sports Science, Juntendo University, 1-1 Hiraka-gakuendai, Inzai 270-1695, Japan; 5Department of Medicine for Orthtopaedics and Motor Organ, Graduate School of Medicine, Juntendo University, 2-1-1 Hongo, Bunkyo-ku, Tokyo 113-8421, Japan; 6Center for Healthy Life Expectancy, Graduate School of Medicine, Juntendo University, 2-1-1 Hongo, Bunkyo-ku, Tokyo 113-8421, Japan; 7Faculty of International Liberal Arts, Juntendo University, 2-1-1 Hongo, Bunkyo-ku, Tokyo 113-8421, Japan

**Keywords:** bone mineral density, osteoporosis, exercise habit, adolescence, older adults

## Abstract

We investigated effects of exercise habits (EHs) in adolescence and old age on osteoporosis prevalence and hip joint and lumbar spine bone mineral density (BMD). Body composition and BMD in 1596 people aged 65–84 years living in Bunkyo-ku, Tokyo, were measured using dual-energy X-ray absorptiometry. We divided participants into four groups by a combination of EHs in adolescence and old age: none in either period (None-None), only in adolescence (Active-None), only in old age (None-Active), and in both periods (Active-Active). Logistic regression models were employed to estimate multivariable-adjusted odds ratios (ORs) for osteoporosis determined by T-score (less than −2.5 SD) using the None-None reference group. In men, the combination of EHs in adolescence and old age was not associated with osteoporosis prevalence. However, the lumbar spine’s BMD was significantly higher in the Active-Active than the None-Active group (*p* = 0.043). In women, the Active-Active group had lower lumbar spine osteoporosis prevalence than the None-None group (OR 0.65; 95% CI, 0.42–1.00, *p* = 0.049). Furthermore, hip BMD was significantly higher in the Active-Active group than in the other three groups (*p* = 0.001). Older women with EHs in adolescence and old age had higher lumbar BMD and lower risk of osteoporosis.

## 1. Introduction

Long-term care in older adults, defined as persons aged 65 years or above, has become a problem in developed countries. In Japan, their number is increasing rapidly: In 2020, Japan had the highest aging rate of 28.7% globally [1]. Owing to the aging population, the number of Japanese individuals requiring long-term care has increased from 2.18 million to 6.06 million in 2000 and 2015, respectively [2,3]. Among the major factors are falls and fractures in Japanese women, which are strongly associated with walking disabilities, institutionalizations, and deaths [4]. Furthermore, a previous study showed that older women have a higher fall rate than their male counterparts [5]; additionally, about half of women and one-third of men experience a fragility fracture during their lifetime [6]. An important risk factor for fracture is osteoporosis, a disease in which bone strength is reduced owing to loss of bone mineral density (BMD) and deterioration of bone quality [7]. Thus, preventing osteoporosis could be an important strategy to reduce the number of people who need long-term care, especially among women.

Physical activity and exercise reduce the risk of osteoporosis, falls, and fractures [8,9,10]. In particular, exercise in adolescence and old age may be beneficial to prevent osteoporosis. For example, BMD greatly increases in adolescence and exercise in adolescence increases peak bone mass (PBM) at around 25 years [11,12]. In addition, a 10% increase in PBM can delay the onset of osteoporosis by 13 years in women [13]. On the other hand, bone loss accelerates from 55 years, especially in women [14], and exercise in old age reduces bone loss [15,16]. Therefore, it can be inferred that a combination of exercise habits in both age periods (e.g., adolescence and old age) may additively increase BMD and prevent osteoporosis. Two previous studies have partly tested this hypothesis. It has been reported that women who exercised in both adolescence and old age have higher forearm BMD than those who did not exercise in either period [17]. Another study showed that men who performed physical activity in both young adulthood (20 to 34 years) and old age (≥65 years) have higher spine and hip joint BMD than men who are physically inactive throughout the lifespan, while there were no significant differences in women [18]. However, the former study did not measure spine and hip joint BMD, which are required for diagnosis of osteoporosis. In addition, the latter study evaluated exercise habits from 20 to 34 years of age, although, BMD greatly increases during adolescence. Finally, and most importantly, it is completely unknown whether the combination of exercise habits in adolescence and old age is associated with lower risk of osteoporosis.

Based on these considerations, we investigated the associations between exercise habits in adolescence and old age and the prevalence of osteoporosis and hip and lumbar spine BMD. We hypothesized that exercise in both adolescence and old age would effectively reduce the prevalence of osteoporosis.

## 2. Methods

### 2.1. Study Design and Participants

This cross-sectional research used baseline data from the Bunkyo Health Study [19]. We recruited individuals aged between 65–84 years living in Bunkyo-Ku, an urban area in Tokyo, Japan. All participants completed the two-day examinations at the Sportology Center from 15 October 2015, to 1 October 2018. Briefly, we evaluated BMD using dual-energy X-ray absorptiometry (DXA). The study protocol was approved by the ethics committee of the Juntendo University in November 2015 (Nos. 2015078, 2016138, 2016131, 2017121, and 2019085). This research was conducted in accordance with the principles outlined in the Declaration of Helsinki. All participants provided written informed consent and were notified that they had the right to withdraw from the trial at any time.

Of the 1629 participants enrolled in the Bunkyo Health Study, we excluded 18 with unavailable data (BMI [*n* = 2], 25-Hydroxyvitamin D [25(OH)D] (*n* = 1), DXA (*n* = 15)). Furthermore, of the remaining 1611 participants, 15 who received corticosteroids for each disease treatment were excluded as well. Finally, 1596 participants (male: 681, female: 915) were included in this analysis (Figure 1); subsequently, they were divided into four groups based on their exercise habits in adolescence and old age. We defined those who answered “yes” to the question “Did you participate in sports club activities when you were in junior high school or high school” as having exercise habits during adolescence, and we also asked them what types of school-based sports club activities they were involved in. On the other hand, those who responded “yes” to the question “Do you currently have exercise habits?” were described as having exercise habits in old age.

### 2.2. BMD and Definitions of Osteoporosis

The BMD of the hip joint (total hip) and lumbar spine (L2–L4) was measured using DXA (Discovery DXA System; Hologic Inc, Marlborough, MA, USA) [20]. Quality assurance for the longitudinal evaluation was performed by calibrating the machine with standardized phantoms. BMD was expressed as standard deviation (SD) units relative to the BMDs of young persons (T-score). According to the World Health Organization criteria, osteoporosis was defined as a T-score less than –2.5 SD [21,22]. In addition, those who received drug treatment for osteoporosis were also defined as having osteoporosis.

### 2.3. Other Measurements

Height was measured within 0.1 cm using a stadiometer (YS-201-P; YAGAMI Inc., Nagoya, Japan) in the upright position. Bodyweight was measured within 0.1 kg using an electronic scale (InBody770; Biospace, Seoul, Korea). Body compositions, including body fat, were measured using DXA. Self-administered questionnaires were employed to determine the following: sex (male or female), age (in years), and smoking status (current and former smoking). Physical activity levels were evaluated using the International Physical Activity Questionnaire (IPAQ) [23,24]. The blood samples were collected in the morning after an overnight fast for biochemical testing. Diabetes was defined as fasting plasma glucose ≥126 mg/dL and/or a 2 h glucose level ≥200 mg/dL after a 75 g oral glucose tolerance test, and hemoglobin A1c ≥6.5%, or currently taking medication for diabetes mellitus. All blood samples were tested at the commissioned clinical laboratory center (SRL Inc., Tokyo, Japan). Dietary intake was assessed using a brief self-administered diet history questionnaire [25,26].

### 2.4. Statistical Analysis

The participants were categorized into the following four groups: Group 1 (no exercise habits in either adolescence or old age (None-None; NN group)), Group 2 (no exercise habit in adolescence, but exercise habit in old age (None-Active; NA group)), Group 3 (exercise habits in adolescence, but not in old age (Active-None; AN group)), and Group 4 (exercise habits in both adolescence and old age (Active-Active; AA group)). Their characteristics were compared using the Kruskal–Wallis and chi-squared tests for continuous and categorical variables, respectively. Continuous variables were reported as medians (interquartile range), while categorical variables were indicated as frequencies (percentages). Logistic regression models were used to estimate the odds ratios (ORs) and the 95% confidence intervals (CIs) for the prevalence of osteoporosis in each group, compared with the NN group. Model 1 was adjusted for age (continuous variable) and BMI (continuous variable). Model 2 was adjusted for the Model 1 covariates plus current and past smoking history (yes or no). Furthermore, Model 3 was adjusted for the Model 2 covariates plus calcium intake (continuous variable), alcohol intake (continuous variable), 25(OH)D level (continuous variable) and diabetes (yes or no). Subsequently, we conducted an analysis of covariance (ANCOVA) to investigate the relationships between the four groups, and BMD. The potential confounders were: age (continuous variable), BMI (continuous variable), calcium intake (continuous variable), alcohol intake (continuous variable), 25(OH)D level (continuous variable), current and past smoking history (yes or no), diabetes (yes or no), taking osteoporosis drugs or estrogens (yes or no). These were reported as the mean and standard errors. Since the prevalence of osteoporosis is relatively different in men and women, we examined the data separately for them. The Statistical Package for the Social Sciences v. 27.0 for Windows (SPSS, Inc., Chicago, IL, USA) was employed to analyze the data. All statistical tests were two-sided, with a 5% significance level.

## 3. Results

### 3.1. Characteristics of the Groups Defined by Exercise Habits in Men and Women

The types of school-based sports club activities the participants were involved in were shown in Appendix A. The characteristics of the four groups described by exercise habits in adolescence and old age of men and women are shown in Table 1 and Table 2, respectively. In the men, the number of participants in the NN group was numerically lower than that in the other three groups (*n* = 86). Body fat was significantly lower in the AA group than in the other groups. Furthermore, the 25(OH)D level in the NA group was greater than that in the AN group, and higher in the AA group than in the NN and AN groups. Physical activity levels in the NA and AA groups were ~2 times higher than in the other two groups, reflecting their current exercise habits. The prevalence of osteoporosis in the hip joint and lumbar spine were 4.7% and 1.2% in the NN group, 4.8% and 2.4% in the NA group, 11.1% and 4.6% in the AN group, and 6.2% and 1.8% in the AA group, respectively, which were considerably lower than those in women (Table 1). ANCOVA revealed that the T-score of BMD in the hip joint was comparable among the groups (Figure 2, *p* = 0.091). In the lumbar spine, the T-score of BMD in the AA group was significantly greater than that in the NA group (*p* = 0.043); moreover, it tended to be higher than that in the NN group (*p* = 0.052), while the BMD values were within the normal range in each group and considerably higher than those in women (Figure 2).

In women (Table 2), the proportion of the NN and NA groups was relatively higher than in men. Body fat was significantly lower in the AA group compared to the NN group. Physical activity levels in the NA and AA groups were ~1.5 times higher than those in the other two groups, which was relatively lower than that in men. Additionally, the prevalence of osteoporosis in the hip joint and joint was significantly greater in the AA group compared to the other three groups (Figure 3, *p* = 0.001 for each). Regarding the female lumbar spine, BMD was slightly higher in the AA group than in the other three groups; however, this difference was not significant (*p* = 0.054).

### 3.2. Exercise Habits and ORs for Osteoporosis

The ORs for osteoporosis in the NA, AN, and AA groups, compared to the NN group, are shown in Table 3. In men, none of the ORs for osteoporosis in the hip joint or lumbar spine were statistically significant. However, those in women regarding the hip joint tended to be lower in the AA group than in the NN group (OR, 0.70; 95% CI, 0.46–1.06, *p* = 0.089) after full adjustment (Model 3). In the lumbar spine, the ORs were significantly lower in the AA group, as compared to the NN group (OR, 0.65; 95% CI, 0.42–1.00, *p* = 0.049) after full adjustment (Model 3).

## 4. Discussion

In the present study, we investigated the additive effects of exercise habits in adolescence and older age on the prevalence of osteoporosis and BMD of the hip joint and lumbar spine in community-dwelling older adults. In men, the T-score of BMD in the hip joint was comparable among the groups. In the lumbar spine, the T-score of BMD in the AA group was significantly higher or tended to be greater than in the NA (*p* = 0.043) and NN (*p* = 0.052) groups, respectively. However, BMD values were within the normal range in each group, and none of the ORs for osteoporosis in the hip joint or lumbar spine were statistically significant. As compared to men, the prevalence of osteoporosis was considerably higher and BMD was significantly lower in women. The T-score of BMD in the hip joint was significantly higher in the AA group than in the other three groups (*p* = 0.001 for each). Although there were no significant differences, we observed a similar trend in BMD in the lumbar spine. The OR for osteoporosis in the lumbar spine (*p* = 0.049) or hip joint (*p* = 0.089) was significantly or tended to be reduced in the AA group, as compared with the NN group (Table 3).

Although this research employed simple qualitative questions to define exercise habits, we observed an association between exercise habits and a lower OR for osteoporosis or a higher BMD in women, suggesting their usefulness in clinical settings. However, in the present study, it is not possible to determine important particularities of the exercise such as types, quantity, intensity, and frequency of exercise on bone health. On the other hand, it is very difficult to validate the questionnaires that ask for details about exercise habits 50–70 years ago in terms of quantity, intensity, and frequency of exercise. Concerning, in Japanese junior and senior high schools, school-based sports club activities called “Bukatsudo” are organized as part of education [27], and participation in school-based sports clubs with other students could be relatively memorable. Therefore, in order to minimize the recall bias and ensure the validity of the questionnaires, we defined exercise habits in adolescent as participation in sports club activities in junior high school and high school. Despite the limitations of this method, we could have chosen a plausible way to identify the exercise habits of adolescents.

The underlying mechanisms of the association between exercise habits and bone health in women remain unclear. Concerning, previous study showed that combined-impact exercise protocols (impact exercise with resistance training) are the best choice to preserve/improve BMD in pre- and postmenopausal women [28]; however, the amount and intensity of exercise in adolescent were not evaluated in the present study. On the other hand, physical activity level evaluated by the IPAQ were consistently higher in those who had existing exercise habits (NA and AA groups) than in those who did not (AN and NN groups). Concerning, it has been suggested that physical activity in old age maintains BMD [29] and exercise habits/physical activity in middle age are related to high BMD later in life [30]. Therefore, we speculate that those who had exercise habits in both periods, adolescence and old age, may have continued exercising between them as well, while the elderly who had exercise habits in either of the two periods may not have. Further studies are required to test this hypothesis.

Compared to women, men had a lower prevalence of osteoporosis and higher T-scores. This result is consistent with previous studies indicating women as having a higher prevalence of osteoporosis than men [31]. Although BMD levels were normal in men, BMD in the male lumbar spine was greater in those who had exercise habits in both adolescence and old age than in those who did not have them in adolescence. Therefore, even in older men with normal BMD, exercise habits in both adolescence and old age may positively associate their BMD in the lumbar spine.

One of the characteristics of the Japanese diet is the low intake of calcium and vitamin D insufficiency [32]. In fact, in the present study, mean calcium intake was below the nutritional recommendation in men (700–750 mg/day) and comparable in women (600–650 mg/day), and mean serum 25(OH)D level was lower than the reference level for vitamin D deficiency (30 nmol/L) in both sexes. Probably due to these dietary characteristics, the prevalence of osteoporosis in Japan is higher than in other developed countries [33]. Therefore, our results may not be applicable to people in other countries.

The current study has several limitations. This cohort included only those individuals who were living in urban areas in Japan. Therefore, we speculate that many of the participants may have been more concerned about their health. As indicated, there is a school-based sports club activity in Japan that is performed in junior and senior high schools as part of education. Thus, it remains unclear whether exercise habits in adolescents in other countries also have similar effects on bone health. This study did not consider physical activity details such as type, frequency, or duration of exercise. According to a systematic review and meta-analysis on type of exercise, basketball players possess a higher BMD than non-athletes, swimming, soccer, and volleyball players, due to the high forces applied during basketball activities, which places a greater burden on the skeletal system [34]. In contrast, sports performed in water such as swimming involve reduced bone loading and may not result in BMD accretion [35]. Additionally, we did not ask exercise habits before adolescents. A previous study showed that exercise habits in childhood is positively associated with PBM [36]. There were a limited number of men in the NN group, and the prevalence of osteoporosis was extremely low; hence, this study may have been statistically underpowered. Since this research was cross-sectional, it was not possible to establish a causal relationship. Therefore, further prospective and interventional studies are needed to clarify the association between exercise habits in adolescence and old age and the incidence of osteoporosis in the latter.

In conclusion, older adults with exercise habits in both adolescence and old age had higher BMD in the lumbar spine and hip joint, in men and women, respectively. Additionally, these were associated with a lower OR for osteoporosis of the lumbar spine in women. Although we could not determine important particularities of the exercise performed, the simple qualitative questions used in this study to define exercise habits may be useful in predicting the risk of osteoporosis in women in clinical settings.

## Figures and Tables

**Figure 1 jcm-10-05968-f001:**
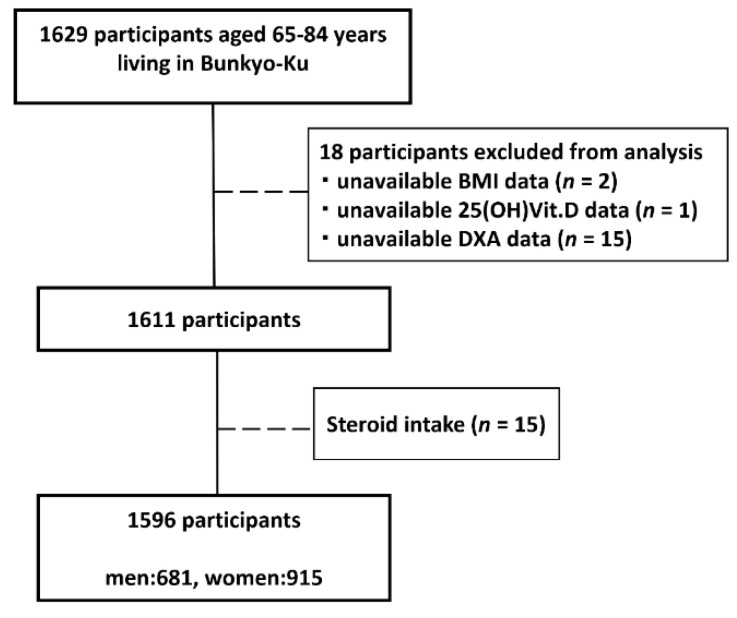
Flowchart of the participants.

**Figure 2 jcm-10-05968-f002:**
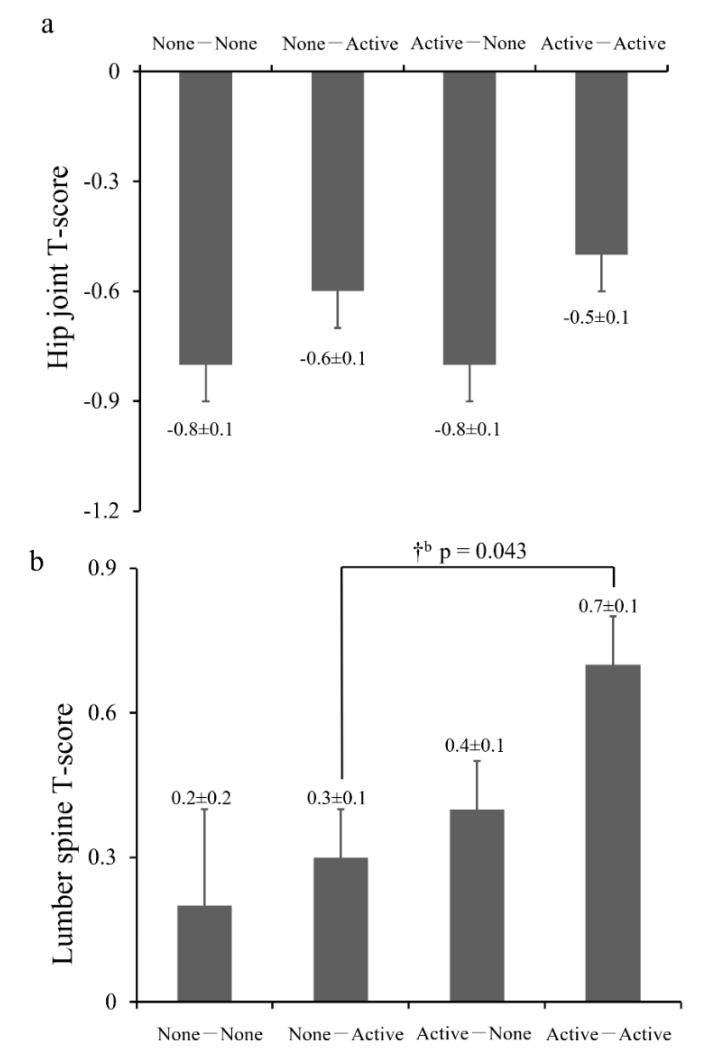
Comparison of bone mineral density (T-score) between the four exercise groups in men. (**a**) Hip joint T-score; (**b**) lumbar spine T-score. Values are the means ± SE. † *p* < 0.05 for significant difference between groups; ^b^: compared to the None-Active group. Adjusted variables: Age, BMI, smoking history (current and past), alcohol intake, calcium intake, 25(OH)D, presence of diabetes, and taking osteoporosis drugs or estrogens.

**Figure 3 jcm-10-05968-f003:**
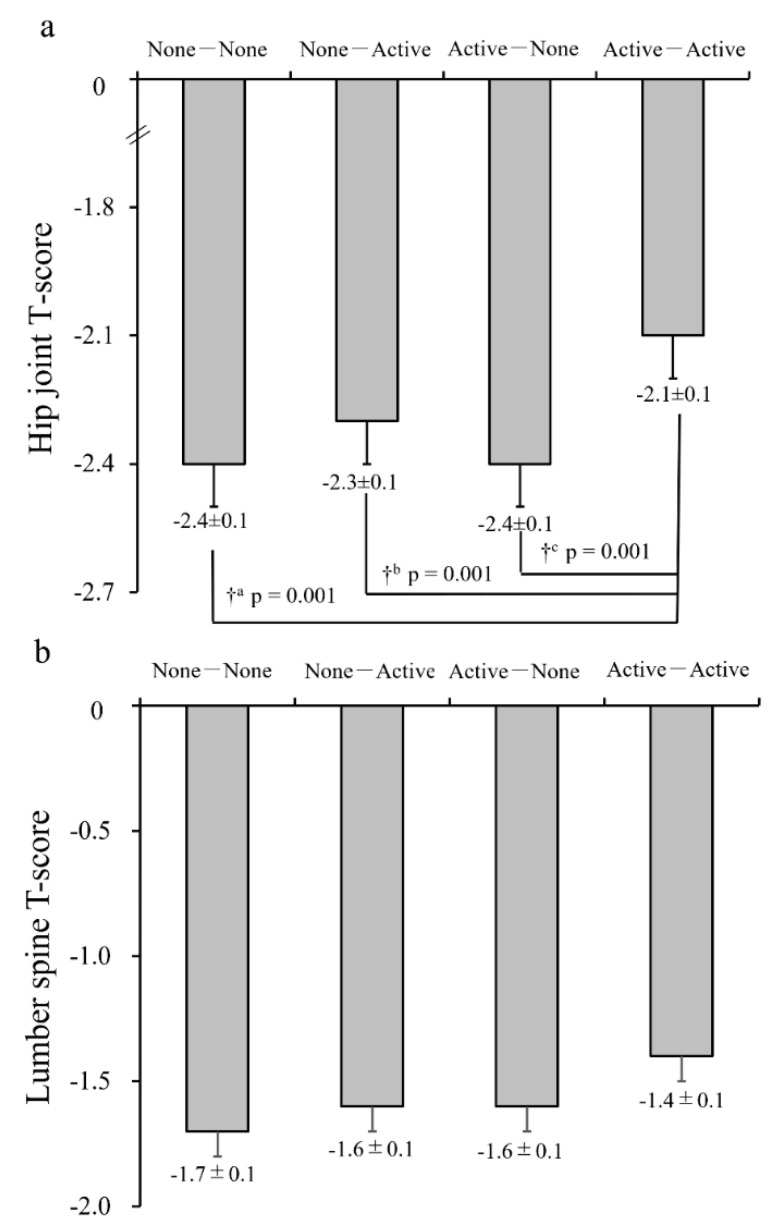
Comparison of the bone mineral density (T-score) between the four exercise groups in women. (**a**) Hip joint T-score; (**b**) lumbar spine T-score. Values are the means ± SE. † *p* < 0.05 for significant difference between groups; ^a^: compared to the None-None group, ^b^: compared to the None-Active group, ^c^: compared to the Active-None group. Adjusted variables: Age, BMI, smoking history (current and past), alcohol intake, calcium intake, 25(OH)D, presence of diabetes, and taking osteoporosis drugs or estrogens.

**Table 1 jcm-10-05968-t001:** Characteristics of the male participants.

	None-None	None-Active	Active-None	Active-Active	*p*-Value
Number (%)	86 (12.6)	167 (24.5)	153 (22.5)	275 (40.1)	
Age (in years)	74 (69–79)	74 (69–78)	72 (68–77)	72 (68–76) ^b^	*p* = 0.032
Height (cm)	164.9 (161.3–168.3)	165.4 (160.6–169.0)	166.0 (162.1–170.1)	166.2 (161.5–170.8)	*p* = 0.143
Bodyweight (kg)	66.9 (62.0–71.3)	63.3 (57.5–68.9) ^a^	66.6 (60.9–71.7) ^b^	64.9 (59.3–71.9)	*p* = 0.003
Body fat (%)	19.1 (16.8–22.6)	18.2 (15.6–21.7)	18.1 (15.4–21.3)	17.3 (14.5–19.8) ^a,b,c^	*p* = 0.000
BMI (kg/m^2^)	24.7 (23.0–26.0)	23.6 (21.7–25.4) ^a^	23.8 (22.3–25.5)	23.5 (21.9–25.5) ^a^	*p* = 0.003
Calcium intake (mg/day)	591 (470–786)	604 (485–844)	633 (477–821)	673 (498–873)	*p* = 0.162
Alcohol intake (g/day)	11.3 (0.00–32.4)	7.59 (0.23–28.2)	14.7 (0.00–39.4)	16.7 (1.39–43.7) ^b^	*p* = 0.005
Physical activity (METs hour/week) *	19.8 (11.2–35.6)	37.4 (23.1–37.2) ^a^	23.1 (13.2–37.2) ^b^	39.8 (23.1–67.2) ^a,c^	*p* = 0.000
Current smoking (*n*/%)	12 (14.0)	13 (7.8)	29 (19.0)	36 (13.1)	*p* = 0.033
Smoking history (*n*/%)	52 (60.5)	116 (69.5)	121 (79.1)	208 (75.6)	*p* = 0.008
Taking osteoporosis medication (*n*/%)	1 (1.2)	1 (0.6)	1 (0.7)	2 (0.7)	*p* = 0.965
Diabetes mellitus (*n*/%)	14 (16.3)	28 (16.8)	31 (20.3)	50 (18.2)	*p* = 0.831
Hypertension (*n*/%)	36 (41.9)	77 (46.1)	79 (53.4)	113 (43.0)	*p* = 0.186
Cancer (*n*/%)	9 (10.5)	26 (15.6)	36 (24.3) ^§^	38 (14.4)	*p* = 0.020
Osteoporosis—Hip Joint (*n*/%)	4 (4.7)	8 (4.8)	17 (11.1)	17 (6.2)	*p* = 0.094
Osteoporosis—Lumbar spine (*n*/%)	1 (1.2)	4 (2.4)	7 (4.6)	5 (1.8)	*p* = 0.276
Hip joint T-score (SD)	−0.8 (−1.6–−0.2)	−0.8 (−1.5–0.3)	−0.9 (−1.8–0.1)	−0.6 (−1.4–0.3)	*p* = 0.102
Lumbar spine T-score (SD)	0.1 (−0.9–1.4)	0.1 (−0.9–−1.1)	0.4 (−0.8–1.6)	0.5 (−0.5–1.8) ^b^	*p* = 0.034
25(OH)D (nmol/L)	18.8 (16.0–22.0)	20.0 (17.0–24.0)	18.0 (15.0–22.2) ^b^	20.0 (17.0–24.0) ^a,c^	*p* = 0.000
Glucose (mg/dL)	94.0 (89.0–104.0)	95.0 (90.0–104.0)	95.5 (90.0–103.0)	96.0 (90.0–105.0)	*p* = 0.661
HbA1c (%)	5.6 (5.4–5.9)	5.7 (5.5–5.7)	5.7 (5.5–6.1)	5.7 (5.5–6.0)	*p* = 0.223
Creatine (mg/dL)	0.70 (0.61–0.81)	0.71 (0.61–0.85)	0.73 (0.64–0.87)	0.71 (0.62–0.86)	*p* = 0.430
eGFR (mL/min)	83.4 (69.6–97.0)	82.1 (68.7–97.5)	79.6 (67.1–91.4)	82.4 (67.1–95.5)	*p* = 0.486

Values are presented as medians (interquartile range). BMI, body mass index; MET, metabolic equivalent of task. ^a^
*p* < 0.05, significant difference compared to the None-None group, ^b^
*p* < 0.05, significant difference compared to the None-Active group, ^c^
*p* < 0.05, significant difference compared to the Active-None group. * Physical activity was measured by IPAQ. ^§^
*p* < 0.05 significantly different for Chi-squared test.

**Table 2 jcm-10-05968-t002:** Characteristics of the female participants.

	None-None	None-Active	Active-None	Active-Active	*p*-Value
Number (%)	159 (17.4)	327 (35.7)	144 (15.7)	285 (31.1)	
Age (in years)	72 (68–76)	74 (69–78)	73 (68–79)	72 (68–77) ^b^	*p* = 0.015
Height (cm)	152.4 (149.2–155.9)	152.0 (148.7–154.8)	153.0 (149.1–156.8)	153.1 (149.3–157.2) ^b^	*p* = 0.006
Bodyweight (kg)	53.4 (47.6–58.4)	51.0 (46.4–55.2) ^a^	53.6 (48.1–61.4) ^b^	52.4 (47.5–56.8)	*p* = 0.001
Body fat (%)	27.2 (23.2–31.4)	25.9 (22.1–29.2)	27.1 (23.4–30.5)	25.8 (22.6–28.6) ^a^	*p* = 0.005
BMI (kg/m^2^)	23.0 (20.6–25.3)	22.2 (20.2–24.1)	23.1 (20.8–25.5)	22.2(20.5–24.3)	*p* = 0.017
Calcium intake (mg/day)	650 (453–823)	725 (552–898) ^a^	637 (488–835)	667(514–930)	*p* = 0.003
Alcohol intake (g/day)	0.00 (0.00–1.17)	0.00 (0.00–2.40)	0.23 (0.00–3.47)	0.47(0.00–6.62)	*p* = 0.048
Physical activity (METs hour/week) *	23.1 (11.6–43.1)	31.8 (18.6–52.2) ^a^	19.8 (9.90–39.6) ^b^	33.1(19.8–57.2) ^a.c^	*p* = 0.000
Current smoking (*n*/%)	5 (3.1)	8 (2.4)	5 (3.5)	12(4.2)	*p* = 0.678
Smoking history (*n*/%)	27 (17.0)	46 (14.0)	28 (19.4)	59(20.7)	*p* = 0.164
Taking osteoporosis medication (*n*/%)	22 (13.8)	60 (18.3)	15 (10.4)	34(11.9)	*p* = 0.124
Diabetes mellitus (*n*/%)	13 (8.2)	27 (8.3)	19 (13.2)	22(7.7)	*p* = 0.256
Hypertension (*n*/%)	74 (46.5)	154 (47.1)	69 (48.6)	122 (43.7)	*p* = 0.771
Cancer (*n*/%)	22 (13.8)	41 (12.5)	12 (16.3)	29 (10.4)	*p* = 0.417
Osteoporosis—Hip Joint (*n*/%)	82 (51.6)	185 (56.5)	82 (56.9)	132(46.3)	*p* = 0.059
Osteoporosis—Lumbar spine (*n*/%)	60 (37.7)	124 (37.9)	53 (36.8)	91(31.9)	*p* = 0.449
Hip joint T-score (SD)	−2.4 (−3.0–−1.7)	−2.4 (−3.1–−1.8)	−2.5 (−3.1–−1.7)	−2.2(−2.8–−1.5) ^b,c^	*p* = 0.004
Lumbar spine T-score (SD)	−1.8 (−2.5–−0.8)	−1.8 (−2.6–−0.8)	−1.5 (−2.5–−0.6)	−1.5(−2.4–−0.5)	*p* = 0.073
25(OH)D (nmol/L)	17.0 (14.1–21.0)	18.0 (15.0–21.0)	17.0 (14.8–21.0)	18.0(15.0–21.5)	*p* = 0.315
Glucose (mg/dL)	96.0 (90.0–107.0)	96.0 (91.0–104.0)	98.0 (91.0–104.0)	98.0 (92.0–104.0)	*p* = 0.326
HbA1c (%)	5.7 (5.5–6.0)	5.7 (5.5–6.0)	5.7 (5.5–6.1)	5.7 (5.5–6.1)	*p* = 0.808
Creatine (mg/dL)	0.7 (0.6–0.82)	0.74 (0.63–0.86)	0.76 (0.65–0.88) ^a^	0.75 (0.66–0.87) ^a^	*p* = 0.014
eGFR (mL/min)	63.1 (51.1–73.5)	57.7 (48.8–69.3)	55.0 (47.9–67.6)	56.9 (48.6–66.7)	*p* = 0.011

Values are presented as medians (interquartile range). BMI, body mass index; MET, metabolic equivalent of task. ^a^
*p* < 0.05, significant difference compared to the None-None group, ^b^
*p* < 0.05, significant difference compared to the None-Active group, ^c^
*p* < 0.05, significant difference compared to the Active-None group. * Physical activity was measured by IPAQ.

**Table 3 jcm-10-05968-t003:** Relationship between exercise habits (four groups) and the prevalence of osteoporosis in the hip joint and lumbar spine.

Hip Joint					
Gender	Group	Crude	Model 1	Model 2	Model 3
Male	None-None	1.00	1.00	1.00	1.00
	None-Active	1.03 (0.30–3.53)	0.83 (0.24–2.90)	0.82 (0.24–2.89)	0.83 (0.24–2.94)
	Active-None	2.56 (0.83–7.88)	2.47 (0.79–7.70)	2.77 (0.88–8.73)	2.70 (0.86–8.53)
	Active-Active	1.35 (0.44–4.13)	1.21 (0.39–3.78)	1.35 (0.43–4.25)	1.37 (0.43–4.36)
Female	None-None	1.00	1.00	1.00	1.00
	None-Active	1.22 (0.84–1.79)	0.97 (0.65–1.45)	0.96 (0.64–1.44)	0.96 (0.64–1.45)
	Active-None	1.24 (0.79–1.95)	1.27 (0.78–2.07)	1.27 (0.78–2.07)	1.28 (0.78–2.09)
	Active-Active	0.81 (0.60–1.20)	0.69 (0.46–1.05)	0.69 (0.46–1.05)	0.70 (0.46–1.06)
**Lumbar Spine**					
**Gender**	**Group**	**Crude**	**Model 1**	**Model 2**	**Model 3**
Male	None-None	1.00	1.00	1.00	1.00
	None-Active	2.09 (0.23–18.96)	1.51 (0.16–14.03)	1.47 (0.16–13.80)	1.27 (0.13–12.27)
	Active-None	4.08 (0.49–33.69)	3.37 (0.40–28.40)	3.13 (0.37–26.57)	3.42 (0.40–29.03)
	Active-Active	1.57 (0.18–13.66)	1.17 (0.13–10.40)	1.10 (0.12–9.88)	0.91 (0.09–8.53)
Female	None-None	1.00	1.00	1.00	1.00
	None-Active	1.02 (0.70–1.51)	0.81 (0.53–1.23)	0.81 (0.53–1.23)	0.79 (0.52–1.21)
	Active-None	0.96 (0.60–1.53)	0.95 (0.58–1.57)	0.95 (0.57–1.56)	0.97 (0.59–1.62)
	Active-Active	0.77 (0.52–1.16)	0.65 (0.42–1.00)	0.65 (0.42–1.00)	0.65 (0.42–1.00)

Data are expressed as odds ratios (95% CIs). Model 1 was adjusted for age and BMI. Model 2 was adjusted for the Model 1 covariates plus current and past smoking history. Model 3 was adjusted for the Model 2 covariates plus calcium intake, alcohol intake, 25(OH)D level, and diabetes.

## Data Availability

Some or all datasets generated and/or analyzed during the current study are not publicly available; however, they can be obtained from the corresponding author upon a reasonable request. The statistical analyses were performed using the Statistical Package for the Social Sciences version 27.0, for Windows (SPSS, Inc., Chicago, IL, USA).

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
