# Peer review of "Associations of Exercise Habits in Adolescence and Old Age with Risk of Osteoporosis in Older Adults: The Bunkyo Health Study"

_jcm, 2021, doi:10.3390/jcm10245968_

Round 1

Reviewer 1 Report

The purpose of this study was to analyze the effects of exercise habits in adolescence and old age on BMD of the hip joint and lumbar spine and the prevalence of osteoporosis. This is a research addressing a current and relevant topic with potential to contribute to the progress of knowledge. It should be highlighted good methodological resources used to process experimental data, although no new experimental design was proposed. Another strength of the present study is the sample used for the analysis: it was used baseline data from the Bunkyo Health Study (1629 people aged 65-84 years old) an interesting prospective cohort study of elderly subjects in an urban community in Japan (doi.org/10.1136/bmjopen-2019-031584). However, there are some methodological questions committing the scientific contribution of this paper. The greatest limitation is related to the procedure used to constitute the experimental groups: participants were divided into four groups based on their answer to two simple questions (“Did you participate in sports club activities when you were in 99 junior high school or high school”; “Do you currently have exercise habits?”).

In this way, it is not possible to determine important particularities of the exercise performed, such as its biomechanical characteristics, intensity, volume and practice time. Such a condition represents a problem, as the type of exercise performed can significantly affect the osteogenic response. The lack of control for these variables compromises the interpretation of the results, making it impossible to support statements presented. Consequently, the conclusions presented are excessively speculative. For example, in the absence of a tool to objectively determine the type of exercise performed by the subjects, the authors state: “there is a traditional sports club activity called “Bukatsudo” in Japan that is performed in junior and senior high schools as part of education. In general, students who belonged to the Bukatsudo practiced three to three and a half hours daily, six days a week, regardless of vacations”. Therefore, it is clear that there is not enough data to support the conclusions presented. 

Author Response

Thank you for checking the details of the manuscript. We have carefully revised the manuscript according to your comments. We believe the quality of our manuscript has improved greatly because of your comments. We have addressed each of the comments and provide point-by-point responses below. The revisions in the main text are highlighted in red.

Point:1

The greatest limitation is related to the procedure used to constitute the experimental groups: participants were divided into four groups based on their answer to two simple questions ("Did you participate in sports club activities when you were in  junior high school or high school"; "Do you currently have exercise habits?"). In this way, it is not possible to determine important particularities of the exercise performed, such as its biomechanical characteristics, intensity, volume and practice time. Such a condition represents a problem, as the type of exercise performed can significantly affect the osteogenic response. The lack of control for these variables compromises the interpretation of the results, making it impossible to support statements presented.

Consequently, the conclusions presented are excessively speculative. For example, in the absence of a tool to objectively determine the type of exercise performed by the subjects, the authors state: "there is a traditional sports club activity called "Bukatsudo" in Japan that is performed in junior and senior high schools as part of education. In general, students who belonged to the Bukatsudo practiced three to three and a half hours daily, six days a week, regardless of vacations". Therefore, it is clear that there is not enough data to support the conclusions presented. 

Response: Thank you for your comment. We agree that it is not possible to determine important particularities of the exercise performed in the present study. On the other hand, it is very difficult to validate the questionnaires that ask for details about exercise habits 50-70 years ago in terms of quantity, intensity, and frequency of exercise. Concerning, in Japanese junior and senior high schools, school-based sports club activities are organized as part of education, and participation in school-based sports clubs with other students could be relatively memorable. Therefore, in order to minimize the recall bias and ensure the validity of the questionnaires, we defined exercise habits in adolescent as participation in sports club activities in junior high school and high school, and simply asked the participants, "Did you participate in sports club activities when you were in junior high school or high school?". We believe that despite the limitations of this method, we have chosen a plausible method to identify the exercise habits in adolescents.

              On the other hand, we also agree that we have excessively speculated on the details of the exercise habits in adolescents. Therefore, based on these limitations and backgrounds, we have carefully revised our discussion and conclusions to avoid excessive speculations including you indicated and to ensure that they are based on objective evidence (Line 245-270, 313-314). On the other hand, we also asked the participants what types of school-based sports club activities they were involved in, so we added them as additional information (Supplementary Table S1, Line100-101, 159-160).

Reviewer 2 Report

The manuscript named "Associations of exercise habits in adolescence and old age with risk of osteoporosis in older adults: The Bunkyo Health Study" investigated the effects of exercise habits in 1,596 participants (male: 681, female: 915) on osteoporosis prevalence in the hip joint and lumbar spine considering bone mineral density (BMD). In general, this study is fascinating and relevant nowadays, mainly in the aging world context, strongly associated with the increased prevalence of non-transmissible chronic diseases such as osteoporosis. Although the aims of the present study are of essential importance, previous studies have already demonstrated the positive effects of physical activity on bone mineral density. The authors pointed a differential of the present study compared to other ones: the relation between life-long physical practice habits and osteoporosis risk, evaluated by BMD in spine and hip joint. The results show that men and women older adults with physical practice habits in adolescence and old age had higher BMD in the lumbar spine and hip joint. In my opinion, the manuscript must be published after some corrections. First, the expression "exercise habits in adolescence or old age" resembles only two points of life that are important to practice exercise: adolescence and old age, when other studies demonstrate that exercise habit is developed since infant age. Another point, no exercise habits in either adolescence or old age are the same as sedentarism. In my opinion, the characterization of the study population may be more detailed about the presence or absence of chronic diseases such as hypertension, diabetes, kidney diseases, cancer. In addition, there is mention about the intake of osteoporosis medication. However, it is well-established that for some menopaused women, hormonal reposition therapy is employed, and it is a relevant bias that could be affecting the results of the present study. The authors related the IPAQ and other biochemical analyses in section material and methods; however, the results are not presented in the manuscript. Does the biochemical testing refer to diabetes determination? Is description for the IPAQ results? Are the physical activity results expressed as METs hour/week? Another point must be clarified in the text: "The OR for osteoporosis in the lumbar spine or hip joint was significantly or tended to be reduced in the AA group, as compared with the NN group "(line 234) in the discussion section. What criteria were used for characterization a "tended to be reduced"? Another relevant point did not observe in the discussion section is the Japanese nutritional habits. There is a possibility that this aspect affects the results in comparison to other regions of the world?

Author Response

Thank you for checking the details of the manuscript. We have carefully revised the manuscript according to your comments. We believe the quality of our manuscript has improved greatly because of your comments. We have addressed each of the comments and provide point-by-point responses below. Our responses are in red font while the revisions in the main text are highlighted in blue.

Point 1: In my opinion, the manuscript must be published after some corrections. First, the expression "exercise habits in adolescence or old age" resembles only two points of life that are important to practice exercise: adolescence and old age, when other studies demonstrate that exercise habit is developed since infant age.

Response 1: Thank you for your comment. We agree with the importance of exercise habits since infant age. Regarding to your suggestion, a previous study showed that exercise habits in childhood is positively associated with peak bone mass (PMID: 11137037). Thus, we have added this information in the limitation section (Line 301-303).

Point 2: Another point, no exercise habits in either adolescence or old age are the same as sedentarism. In my opinion, the characterization of the study population may be more detailed about the presence or absence of chronic diseases such as hypertension, diabetes, kidney diseases, cancer.

Response 2: We appreciate your comment on this point. As you pointed out, it could be possible that subjects with no exercise habits in either adolescence or old age have more comorbidities due to sedentarism. Therefore, we added the data of HbA1c, glucose, creatinine, eGFR, and the prevalence of hypertension and cancer in Table 1 and 2. Those were similar among the groups, but we observed significant difference in prevalence of cancers among the groups in men. Thus, we preliminary added the presence of cancers in men as adjusted variable; however, the data shown in Figure 2 and Table 3 were not significantly changed.

Point 3: In addition, there is mention about the intake of osteoporosis medication. However, it is well-established that for some menopaused women, hormonal reposition therapy is employed, and it is a relevant bias that could be affecting the results of the present study.

Response 3: Thank you for providing these insights. We agree with this comment and we checked the list of medications in study subjects. We found 3 women had either estriol, conjugated estrogens, or estradiol/levonorgestrel, and the latter (estradiol/levonorgestrel) was used for treatment of osteoporosis. Thus, we included these 3 women as taking osteoporosis drugs or estrogens and applied as adjusted variables. In addition, we included one woman who had estradiol/levonorgestrel as a patient with osteoporosis. However, after these corrections, the results shown in Figure 3 and Table 3 were not significantly changed.

Accordingly, we revised the manuscript (Line 150, 188, 194, 212, Figure 3, Table 2 and 3).

Point 4: The authors related the IPAQ and other biochemical analyses in section material and methods; however, the results are not presented in the manuscript. Does the biochemical testing refer to diabetes determination?

Response 4: Thank you for your comment. As you pointed out, we determined diabetes by biochemical testing. In addition, we also added creatinine and eGFR to Table 1 in men and Table 2 in women, as answered in Response 2.

Point 5: Is description for the IPAQ results? Are the physical activity results expressed as METs hour/week?

Response 5: Thank you for your comment. We apologize the data of IPAQ was not clearly described. Physical activity levels presented in Table 1 and 2 were evaluated by IPAQ (METs·hour/week).

Accordingly, we revised the table legends in Table 1 and 2.

Point 6: Another point must be clarified in the text: "The OR for osteoporosis in the lumbar spine or hip joint was significantly or tended to be reduced in the AA group, as compared with the NN group "(line 234) in the discussion section. What criteria were used for characterization a "tended to be reduced"?

Response 6: Thank you for your suggestion. We described as “tended to be” when the p-value was between 0.05 to 0.10. In fact, women tended to have a lower OR (Model 3) in the AA group compared to the NN group in the hip (OR, 0.70; 95% CI, 0.46-1.06, P = 0.089).

Accordingly, we inserted the definition of “trend” (Line 154-155) and P-values in the sentence (Line 232, 240).

Point 7: Another relevant point did not observe in the discussion section is the Japanese nutritional habits. There is a possibility that this aspect affects the results in comparison to other regions of the world?

Response 7: We thank you for this important perspective comment. One of the characteristics of the Japanese diet is the low intake of calcium and vitamin D insufficiency (PMID: 30775488). In fact, in the present study, mean calcium intake was below the nutritional recommendation in men (700-750mg/day) and comparable in women (600-650mg/day), and mean serum 25(OH)D level was lower than the reference level for vitamin D deficiency (30 nmol/L) in both sexes. Probably due to these dietary characteristics, the prevalence of osteoporosis in Japan is higher than in other developed countries (PMID: 24847682). Therefore, our results may not be applicable to people in other countries.

Accordingly, we revised the manuscript (Line 280-287).

Reviewer 3 Report

This is an article that aims to relate exercise habits in adolescence with the prevalence of osteoporosis in old age.
The topic is interesting and pertinent and the study design is adequate, however, it has a methodological limitation: the use of the IPAQ to assess exercise habits. the IPAQ provides a self-report of exercise habits during the week and is not validated to assess habits over many years. What is the validity of evaluating exercise habits in adolescence?
How was this assessment carried out?

Author Response

Thank you for checking the details of the manuscript. We have carefully revised the manuscript according to your comments. We believe the quality of our manuscript has improved greatly because of the reviewer's comments. We have addressed each of the comments and provide point-by-point responses below. Our responses are in red font while the revisions in the main text are highlighted in blue.

Point: This is an article that aims to relate exercise habits in adolescence with the prevalence of osteoporosis in old age. The topic is interesting and pertinent and the study design is adequate, however, it has a methodological limitation: the use of the IPAQ to assess exercise habits. the IPAQ provides a self-report of exercise habits during the week and is not validated to assess habits over many years. What is the validity of evaluating exercise habits in adolescence? How was this assessment carried out?

Response: Thank you for your comment. As you pointed out, the IPAQ is a questionnaire to evaluate current exercise habits, not past exercise history. Therefore, we used IPAQ to evaluate current physical activity only. On the other hand, it is very difficult to validate the questionnaires that ask for details about exercise habits 50-70 years ago in terms of quantity, intensity, and frequency of exercise. Concerning, in Japanese junior and senior high schools, school-based sports club activities are organized as part of education, and participation in school-based sports clubs with other students could be relatively memorable. Therefore, in order to minimize the recall bias and ensure the validity of the questionnaires, we defined exercise habits in adolescent as participation in sports club activities in junior high school and high school, and simply asked the participants, "Did you participate in sports club activities when you were in junior high school or high school?". We believe that despite the limitations of this method, we have chosen a plausible method to identify the exercise habits in adolescents.

              Accordingly, we revised the manuscript to address this issue (Line 245-256).

Round 2

Reviewer 3 Report

This article presents a major problem regarding the evaluation of what kind exercise people did years ago.
Nevertheless, authors clarified some important issues in the revision.

Author Response

Response to Reviewer 3

This article presents a major problem regarding the evaluation of what kind exercise people did years ago.
Nevertheless, authors clarified some important issues in the revision.

Thanks for reviewing our manuscript. 

We are pleased that you have recognized the value of this paper.

Sincerely,

Hiroki Tabata and Yoshifumi Tamura